# The Effect of Pasteurization on the Antioxidant Properties of Human Milk: A Literature Review

**DOI:** 10.3390/antiox10111737

**Published:** 2021-10-29

**Authors:** Hannah G. Juncker, Eliza J. M. Ruhé, George L. Burchell, Chris H. P. van den Akker, Aniko Korosi, Johannes B. van Goudoever, Britt J. van Keulen

**Affiliations:** 1Amsterdam UMC, Emma Children’s Hospital, Department of Pediatrics, Amsterdam Reproduction & Development Research Institute, 1105 AZ Amsterdam, The Netherlands; h.juncker@amsterdamumc.nl (H.G.J.); e.j.ruhe@amsterdamumc.nl (E.J.M.R.); b.j.vankeulen@amsterdamumc.nl (B.J.v.K.); 2Swammerdam Institute for Life Sciences—Center for Neuroscience, University of Amsterdam, 1098 XH Amsterdam, The Netherlands; A.Korosi@uva.nl; 3Medical Library, Vrije Universiteit Amsterdam, 1081 HV Amsterdam, The Netherlands; g.b.burchell@vu.nl; 4Amsterdam UMC, Emma Children’s Hospital, Department of Pediatrics–Neonatology, 1105 AZ Amsterdam, The Netherlands; c.h.vandenakker@amsterdamumc.nl

**Keywords:** donor milk, treatment, Holder pasteurization, oxidative stress, breastmilk, preterm, antioxidant capacity

## Abstract

High rates of oxidative stress are common in preterm born infants and have short- and long-term consequences. The antioxidant properties of human milk limits the consequences of excessive oxidative damage. However, as the mother’s own milk it is not always available, donor milk may be provided as the best alternative. Donor milk needs to be pasteurized before use to ensure safety. Although pasteurization is necessary for safety reasons, it may affect the activity and concentration of several biological factors, including antioxidants. This literature review describes the effect of different pasteurization methods on antioxidant properties of human milk and aims to provide evidence to guide donor milk banks in choosing the best pasteurization method from an antioxidant perspective. The current literature suggests that Holder pasteurization reduces the antioxidant properties of human milk. Alternative pasteurization methods seem promising as less reduction is observed in several studies.

## 1. Introduction

Excessive oxidative stress, the release of ample reactive oxygen species (ROS) in reaction to a broad range of stressors, is common in preterm born infants. They are often exposed to stressors including infections, oxygen supplementation, phototherapy, and parenteral nutrition. ROS are essential at a certain level as they contribute considerably to biological homeostasis and play a role in cell signaling in psychological and pathophysiological processes but can also cause damage to molecules or even cells when present in excess [1]. Disproportionate ROS levels lead to irreversible cell damage, necrosis and apoptosis as a consequence of lipid peroxidation, protein alterations, and DNA oxidation [1]. Accumulation of ROS damage has short- and long-term consequences. As adequate concentrations of antioxidants are absent in preterm born infants, they are highly susceptible to damage from oxidative stress. Moreover, they have an impaired ability to increase the synthesis of antioxidants [2]. Subsequently, preterm infants have an increased risk of developing oxidative stress-related diseases, such as bronchopulmonary dysplasia, necrotizing enterocolitis, periventricular leukomalacia, and retinopathy of prematurity [3,4,5].

Human milk is the gold standard for early life nutrition, especially for preterm born infants. It is not only important for its nutritive value, but also contains numerous different classes of bioactive components, including antioxidant molecules. The antioxidant capacity of human milk refers to the total activity performed by all of the antioxidant molecules present in the milk. Indeed, fresh human milk has a better antioxidant profile than formula [6,7,8,9]. These antioxidant molecules can counteract the detrimental effects of oxidative stress [10]. Mothers who give birth to preterm infants often have a delayed initiation of lactation. Caesarean section, a very common delivery mode for preterm infants, is also associated with a reduction in breastfeeding rates. Consequently, human milk is often not sufficiently available despite the extreme importance of human milk during this stressful period early in life. When the mother’s own milk is not available, donor milk is considered the best alternative [11].

As several viruses, such as human immunodeficiency virus, hepatitis B, cytomegalovirus and bacterial pathogens such as *Escherichia coli*, group B hemolytic streptococci, and other pathogens, could be transmitted via human milk, it is necessary to process donor milk before use to ensure it is safe for consumption by the recipient infant [12,13]. Pasteurization is an effective method to ensure the microbiological quality of the milk in human milk banks [14]. Several methods of pasteurization have been developed including Holder pasteurization (30 min at 62.5 °C), high-temperature short-time pasteurization (15 s at 72 °C), high-pressure treatment (usually 300–800 Megapascal (MPa), <5–10 min) [15]), and ultraviolet-C treatment (200–280 nm [15]). To date, Holder pasteurization is by far the most commonly used method in human milk banks [16]. Although pasteurization is necessary, it may change the composition of human milk. It is known that Holder pasteurization reduces important nutrients in human milk, for example amino acids and vitamins [17]. Moreover, it has been demonstrated that growth of preterm infants receiving pasteurized milk is lower compared to preterm infants receiving unpasteurized milk [18]. It can be hypothesized that pasteurization also changes the activity and concentration of several biological factors, including antioxidants. Currently a clear understanding on the effect of the various pasteurization methods influence human milk antioxidant properties is lacking. Therefore we set out to review the literature describing the effect of different pasteurization or treatment methods on the antioxidant properties of human milk and to provide evidence to guide donor milk banks in choosing the best pasteurization method.

## 2. Materials and Methods

### 2.1. Search Strategy

A systematic search was performed in the databases PubMed, Embase.com, Clarivate Analytics/Web of Science Core Collection, Cumulative Index to Nursing and Allied Health Literature (CINAHL), and the Wiley/Cochrane Library. The timeframe within the databases was from inception to 27 May 2021. The search included keywords and free text terms for (synonyms of) ‘antioxidant’ or ‘oxidative stress’ combined with (synonyms of) ‘human milk’ combined with (synonyms of) ‘pasteurization’. A full overview of the search terms per database can be found in the Appendix A (see Table A1, Table A2, Table A3, Table A4 and Table A5). No limitations on language were applied in the search. Abstracts and conference reports were not excluded.

### 2.2. Data Collection

As shown in Figure 1, 467 records were identified through database searching. After removal of duplicates, 307 titles and abstracts were screened. Twenty-six of these articles were eligible for full text screening, out of which 13 articles were included. Seven additional articles were identified through cross referencing. In total, 20 articles were included in the current literature review. Few articles reported on the same samples. Provided that extra information was presented, all articles were included in the review.

Antioxidant capacity was classified as follows (1) total antioxidant capacity, (2) enzymatic antioxidants, and (3) non-enzymatic antioxidants. All of the articles were sorted according to the categories above and were subsequently divided into the different pasteurization methods. Data were extracted from the articles and systematically summarized. If an article belonged to more than one category, the data were included in all applicable categories.

## 3. Results

### 3.1. Pasteurization Methods

The studies included in this review used several pasteurization methods. Thermal processing is the most commonly used pasteurization method, with Holder pasteurization as the gold standard described most frequently [6,13,14,19,20,21,22,23,24,25,26,27,28,29,30,31,32]. During Holder pasteurization, human milk is heated in a water bath to 62.5 °C for 30 min, and subsequently cooled to approximately 4 °C [33]. Some studies used alternative thermal heating at different temperatures for different durations, for example, high-temperature, short-time pasteurization (72–75 °C for at least 10 s), which was used in three articles [33].

Another pasteurization method is high-pressure treatment, which involves applying hydrostatic high pressure (300–800 MPa) to human milk for 5–10 min [20]. This method was used in four studies [19,20,28,29]. In three of the included articles, thermal and high pressure processing were combined [34].

Other alternative pasteurization methods were reported. (1) Ultraviolet-C treatment is a non-thermal pasteurization method using ultraviolet-c irradiation with a wavelength between 200–280 nm that is generally used in the food industry [21]. (2) During microwave processing, milk samples are treated at 2450 MHz, 800 W until a certain temperature is reached. Subsequently, the human milk samples are cooled [22]. (3) Thermosonication combines heat and ultrasound treatments. Samples are treated in an ultrasound bath at the frequency of 40 KHz and a power of 100 W [14]. The human milk samples are thermosonicated when the water temperature stabilizes (60 °C for 4 min). Subsequently, the samples are cooled to 5 °C [14]. No studies reported the effect of gamma-irradiation on the antioxidant capacity of human milk.

### 3.2. Effects of Pasteurization on the Antioxidant Properties of Human Milk

#### 3.2.1. Total Antioxidant Capacity

Table 1 summarizes the results of the included studies that determined the effect of different pasteurization or treatment methods on the total antioxidant capacity. Total antioxidant capacity (TAC) is a general measure to indicate the level of free radicals scavenged by a test solution that is commonly used to assess the antioxidant status of human milk samples [35]. Different methods to determine the TAC are described in the literature such as: Trolox equivalent antioxidant capacity (TEAC), ferric reducing ability of plasma (FRAP), and cupric reducing antioxidant capacity (CUPRAC) [35]. In the included studies, the most frequently described method was the TEAC assay, which measures the TAC by scavenging 2,2-azinob-bis 3 ethylbenzthiazoline-6-sulfonic acid radicals (ABTS) [14,21,23,24,25]. Moreover, two studies used the DPPH assay, which measures the TAC by scavenging 1,1-diphenyl-2-picrylhydrazyl (DPPH) free radicals [14,23]. Furthermore, some studies used the FRAP method, which measures the extent of the reduction of ferric-tripyridyl triazine (Fe^3+^-TPTZ) to ferrous tripyridyl triazine (Fe^2+^-TPTZ) caused by the antioxidant in question [24].

Studies on the effect of pasteurization on the total antioxidant capacity of human milk show conflicting results. Some studies showed a reduction in total antioxidant capacity after Holder pasteurization [13,14,23] compared with untreated milk, while others showed no influence of Holder pasteurization [6,21,23,24,25].

Other pasteurization techniques, including high-pressure treatment, alternative thermal pasteurization, and ultraviolet-C treatment, did not influence the TAC [14,21,25,26]. However, Ramírez et al. investigated the combination of high-pressure treatment and thermal processing and found an increase in the total antioxidant capacity under the conditions of 400 MPa at −15 °C and at 800 MPa at 30 °C [26]. An increase in total antioxidant capacity was also reported for thermosonication pasteurization (40 kHz, 100 W, and 60 °C for 4 min).

#### 3.2.2. Enzymatic Antioxidants

Table 2 summarizes the results of the included studies that determined the effect of different pasteurization or treatment methods on the enzymatic antioxidants. 

*Superoxide dismutase:* superoxide dismutase is essential for the elimination of superoxide radicals in cells. It catalyzes the reaction of superoxide into hydrogen peroxide and dioxygen [36]. The produced hydrogen peroxide will subsequently be converted into water through other antioxidant enzymes such as catalase and glutathione peroxidase [36]. Marinković et al. showed that Holder pasteurization caused a decrease in superoxide dismutase activity [6]. Others showed a reduction following heating from 70 °C upwards only [22]. High-pressure treatment increased the activity of superoxide dismutase by 57% at 193 MPa [25]. Microwave heating also led to a significant increase in superoxide dismutase activity (10%, 21%, and 34% at 62.5, 66, and 70 °C for 1 min, respectively) [22].

*Catalase:* catalase is a dismutase enzyme as well and counters the detrimental action of free radicals in the cell. It plays an essential role in the degradation of hydrogen peroxide into water and oxygen [36]. Catalase activity was reduced by approximately 60% after Holder pasteurization [21,22]. Following alternative heat treatment (70 °C for 30 min), a 66% decrease in activity was observed [22]. Microwave heating at 62.5, 66, and 70 °C for 1 min also reduced catalase activity in human milk by 34%, 42%, and 38%, respectively [22]. Ultraviolet-C treatment did not affect catalase activity [21].

*Glutathione peroxidase:* glutathione peroxidase plays an important role in catalyzing the decomposition of hydrogen peroxide in cells. Furthermore, glutathione peroxidase protects the cell against oxidative stress through the decomposition of lipid hydroperoxides [36]. Several studies have investigated the effect of pasteurization on glutathione peroxidase activity. Holder pasteurization showed a reduction in glutathione peroxidase in human milk, ranging from 23% to 70% [6,13,22,27]. Alternative heating methods showed a reduction from 41% to 56% [22]. After high-pressure treatment, glutathione peroxidase activity in human milk decreased by 62% [13]. Microwave heating for 1 min at temperatures of 62.5, 66, and 70 °C led to a 38%, 38%, and 53% decrease in glutathione peroxidase activity, respectively [22].

#### 3.2.3. Non-Enzymatic Antioxidants

The antioxidant properties of human milk are also affected by non-enzymatic antioxidants, including glutathione, vitamin C, E, A, and other agents. Table 3 summarizes the results of the included studies that determined the effect of different pasteurization or treatment methods on the non-enzymatic antioxidants.

*Glutathione:* Glutathione is an important antioxidant in human milk, as it plays a role in the deactivation of oxygen-derived free radicals and the elimination of toxins, carcinogens, and malonic dialdehyde [37]. Silvestre, et al. investigated the effects of Holder pasteurization and high-temperature, short-time pasteurization, and found that Holder, but not HTST pasteurization reduced glutathione concentrations in human milk with 46% [13].

*Vitamin C:* Vitamin C and its isolate ascorbic acid are natural antioxidant components present in human milk. They have efficacious antioxidant function due to its ability to donate electrons and thus protecting important biomolecules [38]. Six studies showed that Holder pasteurization decreased vitamin C between 12% to 40% and ascorbic acid between 16% to 26% [6,21,28,29,30,31]. Alternative thermal pasteurization (100 °C, 5 min) reduced vitamin C and ascorbic acid more severely compared to Holder pasteurization, with a decrease of 29% for vitamin C and 41% for ascorbic acid [30]. Studies on the effect of high-pressure treatment on vitamin C and ascorbic acid report conflicting results. One study found that high-pressure treatment (193 MPa, −20 °C) led to a decrease of approximately 11% in ascorbic acid [28], while another study (400–600 MPa, 5 min, 12 °C) found that it did not alter ascorbic acid concentration in human milk [29]. In addition, vitamin C levels were not influenced by high-pressure treatment [28,29]. With respect to ultraviolet-C treatment, one study found that pasteurization with a radiation dose of 173–740 J/L reduced vitamin C levels by 15% to 35% [21].

*Vitamin E:* Vitamin E can be divided into four different tocopherols (α-, β-, γ-, and δ- tocopherol) and four different tocotrienols. Vitamin E has an important antioxidant function as it protects polyunsaturated fatty acids and other substances from peroxidation [20]. The most abundant tocopherol in human milk is α-tocopherol, which is the most biologically active form of vitamin E [36]. Several studies investigated the effect of pasteurization on tocopherols in human milk. Four studies investigated the effect of Holder pasteurization on α-tocopherol: two showed a reduction [20,30], while the other two showed no effect [29,31]. One study showed that heating at 100 °C for 5 min also reduced the α-tocopherol activity in human milk [30]. High-pressure treatment reduced α-tocopherol between 21–27% at 600 MPa for 3–6 min [20], while another study showed no effect of high-pressure treatment at 400, 500, and 600 MPa. A combination of thermal treatment at 65 and 80 °C at any pressure showed a reduction in α-tocopherol [34].

The studies investigating the effect of pasteurization on γ- and δ-tocopherol also showed conflicting results. Three studies investigated the effect of Holder pasteurization on γ-tocopherol, of which two showed a reduction between 13–47% [20,30] and one showed no effect [29]. Two studies investigated the effect of Holder pasteurization on δ-tocopherol [20], with one showing a reduction of 33% and the other no effect [29]. After alternative thermal pasteurization (100 °C for 5 min) a reduction was found in γ-tocopherol [30]. Two studies investigated the effect of high pressure on γ- and δ-tocopherol, of which one study showed a reduction only at a specific pressure and the other showed no effect [20,29]. A combination of thermal and high pressure treatment also reduced both γ- and δ-tocopherol in human milk [34].

*Vitamin A:* Vitamin A consists of provitamin A (carotenoids: α-carotene and β-carotene) and non-provitamin A (lutein, zeaxanthin, and lycopene). Provitamin A is the most important for vitamin A activity, however, non-provitamin A also has antioxidant properties [39]. Several studies investigated the effect of pasteurization on provitamin A. No effect of Holder pasteurization on vitamin A or β-carotene specifically was found [14,31]. High-pressure treatment did not influence β-carotene activity, and microwave heating (35–40 °C, 15–30 s) did not influence α- or β-carotene [19,40]. Moreover, no differences in vitamin A stability were found after the thermosonication pasteurization of human milk samples [14].

Several studies investigated the effect of pasteurization on non-provitamin A. Holder pasteurization reduced lutein and zeaxanthin by 16%, while it increased lycopene by 9% [19]. High-pressure treatment (100–600 MPa) reduced the concentrations of lutein and zeaxanthin by 40% to 60%, while it increased the concentration of lycopene by 6% to 14% [19]. After microwave heating, no effects on the concentrations of lycopene and lutein were observed.

*Other antioxidants:* Some trace elements contribute to the antioxidant function of human milk. For example, zinc and copper are cofactors in the superoxide dismutase reaction [32]. Selenium also contributes to antioxidant properties as it is part of the antioxidant selenoprotein enzyme that protects the cell against free radicals [32]. Moreover, selenium is a component of glutathione peroxidase [32]. Holder pasteurization did not affect zinc, copper, or selenium levels in human milk [32].

**Table 1 antioxidants-10-01737-t001:** Results of different pasteurization methods on antioxidant properties in human milk: total antioxidant capacity.

Antioxidant Components	Pasteurized Milk Samples	Holder Pasteurization	Alternative Thermal Pasteurization	Ultraviolet-C Treatment	Thermosonication	References
ABTS	N = 8	↓ (30.5%)	-	-	-	[23]
	N = 21 *	=	-	-	-	[24]
	N = 20	=	-	55, 173, 355, 544 and 740 J/L): = (N = 18)	-	[21]
	N = 10	↓ (% n.r.)	-	-	↑ (% n.r.)	[14]
	N = 7	=	20 °C, 193 MPa: =	-	-	[25]
DPPH	N = 8	=	-	-	-	[23]
	N = 10	↓ (% n.r.)	-	-	↑ (% n.r.)	[14]
FRAP	N = 21 *	=	-	-	-	[24]
ORAC	N = 10	=	-	-	-	[6]
ORP	N = 10	=	-	-	-	[6]
Unknown	N = 31	↓ (67%)	75 °C, 15 s: =	-	-	[13]
	N = 3	-	−15, 0, 10, 30 and 50 °C combined with 200, 400, 600 and 800 MPa for 1 s: = −15 °C, 400 MPa and −30 °C, 800 MPa: ↑	-	-	[26]

Abbreviations: MPa = Megapascal; J/L = Joule per liter; ABTS = 2,2′-azino-bis (3-ethylbenzothiazoline-6-sulfonzuur); DPPH = 2,2-difenyl-1-picrylhydrazyl; FRAP = ferric reducing ability of plasma; ORAC = oxygen radical absorbance capacity; ORP = oxidation-reduction potential; n.r. = not reported. The N indicates the number of pasteurized human milk samples. The effect of pasteurization is displayed compared with untreated human milk samples. * In this study, pasteurized human milk samples were compared with human milk samples from a control group instead of within-person comparisons. The symbols in this table should be read as: ↑ increase, = no effect, ↓ decrease.

**Table 2 antioxidants-10-01737-t002:** Results of different pasteurization methods on antioxidant properties in human milk: enzymatic antioxidants.

Antioxidant Component	Pasteurized Milk Samples	Holder Pasteurization	Alternative Thermal Pasteurization	Microwave Heating	Ultraviolet-C Treatment	References
Superoxide dismutase	N = 10	↓ (% n.r.)	-	-	-	[6]
	N = 10	=	66 °C, 0–30 min: = 70 °C, 0–20 min: = 70 °C, 30 min: ↓ (35%)	70 °C 1 min: ↑ (34%)	-	[22]
	N = 7	-	<0 °C, 193 MPa: ↑ (57%)	-	-	[25]
Catalase	N = 10	↓ (57%)	66 °C, 0–30 min: ↓ (59.7–81.9%) 70 °C, 0–30 min: ↓ (58.4–86.9%)	62.5 °C 1–10 min: ↓ (33–39%) 66 °C 1–10 min: ↓ (39–48%) 70 °C 1–10 min: ↓ (38–52%)	-	[22]
	N = 8	↓ (60%)	-	-	85–740 J/L: =	[21]
Glutathione Peroxidase	N = 17	↓ (63%)	75 °C, 15s: ↓ (62%)	-	-	[13]
	N = 10	↓ (% n.r.)	-	-	-	[6]
	N = 21	↓ (23–70%)	-	-	-	[27]
	N = 10	↓ (42%)	66 °C, 0–30 min: ↓ (41.1–45.8%) 70 °C, 0–30 min: ↓ (44.4–56.1%)	62.5 °C 1–10 min: ↓ (11–44%) 66 °C 1–10 min: ↓ (28–42%) 70 °C 1–10 min: ↓ (31–53%)	-	[22]

Abbreviations: MPa = Megapascal; J/L = Joule per liter; n.r. = not reported. The N indicates the number of pasteurized human milk samples. The effect of pasteurization is displayed compared with untreated human milk samples. The symbols in this table should be read as: ↑ increase, = no effect, ↓ decrease.

**Table 3 antioxidants-10-01737-t003:** Results of different pasteurization methods on antioxidant properties in human milk: non-enzymatic antioxidants.

Antioxidant Components	Pasteurized Milk Samples	Holder Pasteurization	Alternative Thermal Pasteurization	High Pressure Treatment	Microwave Heating	Ultraviolet-C Treatment	Thermosonication	References
Glutathione	N = 31	↓ (46%)	75 °C, 15 s: =	-	-	-	-	[13]
Vitamin C	N = 7	↓ (35%)	-	−20 °C, 193 MPA: =	-	-	-	[28]
	N = 10	↓ (19.9%)	-	400, 500 and 600 MPa: =	-	-	-	[29]
	N = 5	↓ (36%)	-	-	-	-	-	[31]
	N = 11	↓ (38.4%)	-	-	-	85 J/L: = 173–740 J/L: ↓ (15–35%)	-	[21]
	N = 10	↓ (12%)	100 °C, 5 min: ↓ (29%)	-	-	-	-	[30]
Ascorbic acid	N = 7	↓ (24%)	-	−20 °C, 193 MPA: ↓	-	-	-	[28]
	N = 10	↓ (16.2%)	-	400, 500 and 600 MPa: =	-	-	-	[29]
	N = 10	↓ (% n.r.)	-	-	-	-	-	[6]
	N = 10	↓ (26%)	100 °C, 5 min: ↓ (41%)				-	[30]
Vitamin E: α-Tocopherol	N = 3	↓ (25%)	-	400 MPa, 3/6 min: = 600 MPa, 3/6 min: ↓ (21%, 27%)	-	-	-	[20]
	N = 9	=	-	-	-	-	-	[31]
	N = 10	=	-	400, 500 and 600 MPa: =	-	-	-	[29]
	N = 10	↓ (13–17%)	100 °C, 5 min: ↓ (32–34%)	-	-	-	-	[30]
	N = 6	-	50 °C, 300 MPa, 600 MPa, 900 MPa: = 65/80 °C, any pressure: ↓	-	-	-	-	[34]
Vitamin E: γ- Tocopherol	N = 3	↓ (47%)	-	400, 600 MPa, 3/6 min: ↓ (26–47%)		-	-	[20]
	N = 10	=	-	400, 500 and 600 MPa: =	-	-	-	[29]
	N = 10	↓ (13–17%)	100 °C, 5 min: ↓ (32–34%)	-	-	-	-	[30]
	N = 6		50, 65 and 80 °C, 300/600 MPa: ↓ 50 °C, 900 MPa: = 65 and 85 °C, 900 MPa: ↓	-	-	-	-	[34]
Vitamin E: δ- Tocopherol	N = 3	↓ (33%)	-	400 MPa, 3–6 min: = 600 MPa, 3–6 min: ↓ (25, 33%)	-	-	-	[20]
	N = 10	=	-	at 400, 500 and 600 MPa: =	-	-	-	[29]
	N = 6	-	50, 65 and 80 °C, 300/600 MPa: ↓ 50 °C, 900 MPa: = 65 and 85 °C, 900 MPa: ↓	-	-	-	-	[34]
Vitamine A:	N = 9	=	-	-	-	-	-	[31]
	N = 10	=	-	-	-	-	=	[14]
Vitamine A: β-carotene	N = 18–24	=	-	=	-	-	-	[19]
	N = 30	-	-	-	35–40 °C, 15–30s: =	-	-	[40]
Vitamine A: α-Carotene	N = 30	-	-	-	35–40 °C, 15–30 s: =	-	-	[40]
Vitamine A: Lutein + Zeaxanthin	N = 18–24	↓ (15.8%)	-	600 MPa: ↓ (60.2%) 200 + 400 MPa: ↓ (47.1%) 200 + 600 MPa: ↓ (57.5%) 100 + 600 MPa: ↓ (40%) 450 MPa: ↓ (57.6%)	-	-	-	[19]
Vitamine A: Lycopene:	N = 30	-	-	-	35–40 °C, 15–30 s: =	-	-	[40]
Vitamine A: Luteine	N = 30	-	-	-	35–40 °C: 15–30 s: =	-	-	[40]
Trace elements	N = 16	Zinc: = Copper: = Selenium: =	-	-	-	-	-	[32]

Abbreviations: MPa = megapascal; J/L = joule per liter; n.r. = not reported. The N indicates the number of pasteurized human milk samples. The effect of pasteurization is displayed compared with untreated human milk samples. The symbols in this table should be read as: = no effect, ↓ decrease.

## 4. Discussion

This literature review summarized the current evidence on the influence of pasteurization on the antioxidant properties of human milk and compared different pasteurization methods. Overall, several studies have been conducted on investigating different human milk components and these studies suggest that Holder pasteurization reduces the antioxidant properties of human milk. Alternative pasteurization methods seem promising as less reduction is observed in several studies.

Currently, the most common pasteurization method is Holder pasteurization, in which milk is exposed to a temperature of approximately 62.5 °C for at least 30 min [41]. Most studies investigating the effect of Holder pasteurization on the total antioxidant capacity of human milk showed no effect, whereas others, including the study with the largest sample size (13), did show a reduction of up to 67% of the total antioxidant capacity. The differences between studies could partly be due to the different methods used to measure the total antioxidant capacity. When evaluating the separate antioxidant components that contribute to the total antioxidant capacity of human milk, in most studies, a reduction was observed after Holder pasteurization. In particular, catalase, glutathione peroxidase, glutathione, vitamin C, and ascorbic acid decreased after Holder pasteurization in all of those studies. Studies on the effect of Holder pasteurization on the other antioxidant components of human milk were controversial in their results.

An explanation for the reduction of antioxidant activity in Holder-pasteurized donor milk might be the denaturation of proteins during the heating process [42]. Holder pasteurization is recommended by all international human milk bank guidelines. However, as this process seems to reduce some of the beneficial effects of human milk, safe alternatives should be considered. To date, studies on the effects of other pasteurization techniques on human milk components and, specifically, antioxidant function, are scarce. The limited results demonstrate that alternative pasteurization methods including high-pressure treatment, microwave heating, and ultraviolet-C treatment seem promising and cause less of a reduction in the antioxidant components, especially enzymatic agents. These results are in line with the fact that enzymes denature during the heating process. Moreover, non-enzymatic agents such as vitamins were also less affected by alternative pasteurization methods, which might be due to the size of the molecules, as small molecules are less affected by, for example, high pressure [29]. Remarkably, some studies even show an increase in the antioxidant components of human milk after alternative pasteurization methods. For example, after high-pressure treatment and microwave heating an increase in superoxide dismutase was found. A possible explanation might be that neutrophils in human milk are destroyed under high pressure, causing a release of superoxide dismutase from these cells [34]. During microwave heating, the energy transfer between the electromagnetic field and protein domains might change the enzymatic properties and subsequently increase the reactivity of enzymes, for example, superoxide dismutase [43]. Altogether, this confirms the main concerns about the Holder pasteurization of donor milk and the need for future strategies for which the above-described pasteurization methods seem promising.

As preterm infants are especially susceptible to damage from oxidative stress, the antioxidant properties of human milk are extremely important for this specific group. These infants receive donor milk relatively frequently in the first days of life, when the amount of oxidative stress is at its highest. Therefore, knowledge on the effect of pasteurization methods on the antioxidant activity is important. Moreover, it has been demonstrated that preterm infants receiving pasteurized human milk have poorer growth and developmental outcomes compared to infants receiving unpasteurized human milk [18,44]. This is most likely due to the effect of pasteurization on multiple aspects of human milk and not only on the effect on antioxidants. Thus, it is important to investigate the effect of pasteurization on other human milk components as well, for instance the effect of pasteurization on nutrients, immunological components and the human milk microbiome.

To draw conclusions and enable unbiased, quantitative comparisons of the effect of pasteurization methods on the antioxidant properties of human milk to guide and advise human donor milk banks, future studies should take the following aspects into account. First, study protocols and procedures should be standardized (e.g., sample origin, storage conditions, methods for measuring the TAC). Second, studies should compare different pasteurization techniques within the same human milk samples as antioxidant capacity of human milk is influenced by several other factors, for example, maternal diet [45]. Third, it appears that some antioxidants were investigated more extensively than others. For instance, some short chain fatty acids and amino acids are, next to their nutritional value, also known for their antioxidant properties and were not described in this review [46,47]. Fourth, investigating the functionality of antioxidant components in human milk should be preferred over measuring the concentration. Moreover, studies investigating oxidative stress-related outcomes between preterm infants receiving unpasteurized milk and donor milk treated with different pasteurization methods are currently lacking. Randomized controlled trials should be setup to investigate the effect of pasteurization on these outcomes to draw conclusions on the clinical relevance and to optimize current human milk bank pasteurization guidelines.

## 5. Conclusions

As many human milk components remain relatively unaffected by pasteurization and cow’s milk-based formula does not contain many bioactive components, donor milk is still considered the best alternative for a mother’s milk. Research on the effects of pasteurization method on the antioxidant properties of human milk is scarce. In general, Holder pasteurization seems to reduce the antioxidant properties of human milk, specifically, the activity of certain antioxidant molecules. Whether this reduction is clinically relevant remains to be determined. Alternative pasteurization methods seem promising as less reduction in the TAC was observed in several studies. More research is necessary to improve knowledge on the effect of different pasteurization methods on human milk antioxidants to guide human milk banks in their pasteurization processes to optimize early-life nutrition and, thereby also, the health outcomes of preterm infants.

## Figures and Tables

**Figure 1 antioxidants-10-01737-f001:**
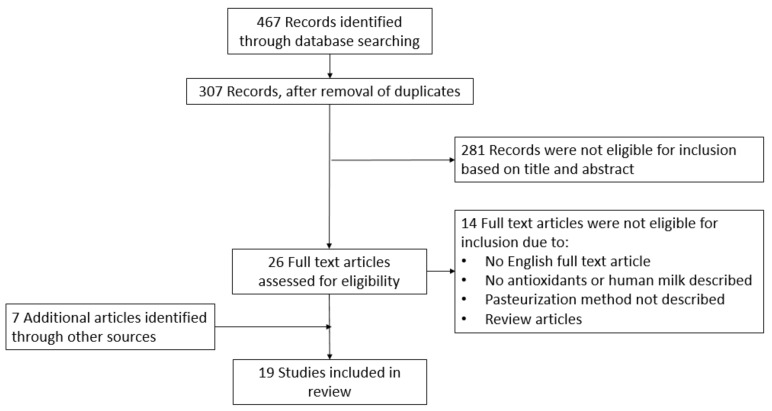
This flowchart presents the different phases of the review, according to the PRISMA-statement.

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
