# Peer review of "The Effect of Pasteurization on the Antioxidant Properties of Human Milk: A Literature Review"

_antioxidants, 2021, doi:10.3390/antiox10111737_

Round 1

Reviewer 1 Report

The present review of Junker and colleagues is a comprehensive review about the effect of pasteurization on antioxidant properties of human milk. In my opinion the present review is well organized and below I add only some suggestions to improve the manuscript.

Line 80 “and conducted by”: please move this information in the “Author Contributions” correct section.

Table 1: I found this table dispersive and not well organized, with too much N.A. I suggest to discuss and split the studies depending on the pasteurization method in addition to antioxidant component. Moreover, I suggest to add a figure or table in which the major findings about the topic are summarized.

Lines 299-300 and 324: please consider that pasteurization can also compromise other important molecules in the human milk, and no less important human milk microbiota. Please add this specification in the text.

Line 310 ‘some were not considered at all’: please list them.

Reviewer 2 Report

The Authors attempted to review available literature on the influence of different pasteurization methods on the antioxidant properties of human milk. The topic is relevant since preterm infants are especially susceptible to damage from oxidative stress, and the antioxidant properties of human milk are important for this specific group. Expanding our knowledge on the effect of different pasteurization methods on the antioxidant activity of human milk could be relevant to drive human milk banks in the pasteurization of donor milk.

The study results confirmed the current knowledge in the field: the pasteurization negatively influence the antioxidant activities of human milk. 

The manuscript is well organized and written. the topic is interesting and relevant to the aims and scopes of antioxidant.
However, I would suggest to modify the introduction and/or add a paragraph.
The authors, indeed, in the introduction section deals with preterm birth and the importance of milkbut no data are reported in the manuscript about the difference between pasteurized and unpasteurized milk in terms of nutrition of preterm chidren.

Overall, the methodology of the research is sound. The results are well presented.
Main weaknesses are related to the many repetitions and redundancies along the text. In addition, the Authors were not able to provide data on the clinical relevancy of the reduced antioxidant activity of the pasteurized human milk. Adding more data on this aspect could be relevant or at least discuss the problem as limitations and perspectives. The potential role of also other human milk components with a strong antioxidant activity such as short chain fatty acids could be also discussed. Finally, a well-structured proposed agenda for the future research in this area could be helpful, containing also discussion as above indicated.
